# Selecting High-Performance Gold Nanorods for Photothermal Conversion

**DOI:** 10.3390/nano12234188

**Published:** 2022-11-25

**Authors:** Túlio de L. Pedrosa, Sajid Farooq, Renato E. de Araujo

**Affiliations:** 1Laboratory of Biomedical Optics and Imaging, Federal University of Pernambuco, Recife 50740-540, Brazil; 2Center for Lasers and Applications, Instituto de Pesquisas Energeticas e Nucleares, IPEN—CNEN, Sao Paulo 05508-000, Brazil

**Keywords:** photothermal conversion, metallic nanoparticles, thermoplasmonics

## Abstract

In this work, we establish a new paradigm on identifying optimal arbitrarily shaped metallic nanostructures for photothermal applications. Crucial thermo-optical parameters that rule plasmonic heating are appraised, exploring a nanoparticle size-dependence approach. Our results indicate two distinct figures of merit for the optimization of metallic nanoheaters, under both non-cumulative femtosecond and continuum laser excitation. As a case study, gold nanorods are evaluated for infrared photothermal conversion in water, and the influence of the particle length and diameter are depicted. For non-cumulative femtosecond pulses, efficient photothermal conversion is observed for gold nanorods of small volumes. For continuous wave (CW) excitation at 800 nm and 1064 nm, the optimal gold nanorod dimensions (in water) are, respectively, 90 × 25nm and 150 × 30 nm. Figure of Merit (FoM) variations up to 700% were found considering structures with the same peak wavelength. The effect of collective heating is also appraised. The designing of high-performance plasmonic nanoparticles, based on quantifying FoM, allows a rational use of nanoheaters for localized photothermal applications.

## 1. Introduction

In recent years, the use of metallic nanostructures for delivery of heat in many photothermal applications have been the subject of numerous studies, ranging from solar harvesting, where plasmonic nanostructures have been used to increase the efficiency of solar absorption [1,2,3,4], to biomedical endeavors, where tissue temperature control has been widely used for therapeutic applications in several areas of medicine such as oncology, physiotherapy, urology, cardiology and ophthalmology [5,6]. Photothermal Therapy (PTT), for instance, is a technique that relies on induction of cell damage by light absorption in a target tissue [7,8,9]. High-performance PTT is achieved by exploring nanoheaters as metallic nanoparticles (NP) for localized photothermal conversion. Among applications where a single or few NP are explored, laser-mediated microbubble generation stands out [10,11]. Microbubble generation with the aid of metallic NPs has been employed in cell membrane photoporation for intracellular delivery of molecules [12,13,14]. Furthermore, ultrafast photothermal processes, around NP, have also been proposed as an alternative optical switching method in communication devices [15].

In such cases, the collective coherent oscillation of conduction electrons on metallic nanoparticles, known as Localized Surface Plasmon Resonance (LSPR), induced by electromagnetic waves, leads to the enhancement of the NP absorption and scattering properties [16]. When radiation strikes, a metallic nanostructure, inducing the LSPR, the absorbed light energy promotes electrons to an excited state above the metal Fermi level, changing the conducting band population distribution, as depicted in Figure 1a. Excited electron–electron scattering will promote the thermalization of carriers, as shown in Figure 1b. The interaction of hot-electrons with the nanocrystal lattice will then take place after a few ps, which corresponds to the internal electron–phonon characteristic relaxation time ($\tau_{ep}$). As described in Figure 1c, only later thermal energy is released to the surrounding medium, mainly by a conductive contribution, causing the NP to act as a nanoheater. The thermal relaxation characteristic time of a metallic NP is associated to its size and is described by [17]:(1)$$\tau_{d}=\frac{\rho_{Au}c_{pAu}}{3\kappa_{m}}R_{eq}^{2},$$
where $R_{eq}$ is the equivalent radius of a sphere the with same volume of the metallic NP (with an arbitrarily shaped), $\rho_{Au}$ and $c_{pAu}$ are, respectively, the gold density and specific heat capacity, and $\kappa_{m}$ is the thermal conductivity of the surrounding medium. In water, the metallic NP $\tau_{d}$ can reach up to a few nanoseconds.

For biomedical applications in particular, the use of infrared (IR) light sources at the biological window allows deep tissue (few centimeters range) treatment due to minimal absorption and scattering processes. Two spectral bands are defined as ideal for deep tissue optical therapy: (i) the first biological window that extends from 700 nm to 980 nm; and (ii) the second biological window, which extends from 1000 nm to 1400 nm [18]. Many groups have been working on different methodologies to prepare NPs of various structures, leading to high optical cross-sections and tunable plasmon spectrum at the near-infrared (NIR) band. High IR absorption cross-section values can be achieved by NP with various shapes, such as rods [19], shells [20] and cages [7,21,22]. In particular, metallic nanorods (NR) have outstanding tunability, making them a strong candidate in biomedical applications. However, infrared optical therapies raise concerns regarding accessibility from a clinical standpoint. For instance, laser exposition of human skin at 1064 nm is limited to 100 mJ/cm^2^ for laser pulses shorter than 100 ns and 1 W/cm^2^ for continuous wave (CW) illumination [23]. Therefore, in photothermal therapy assisted by plasmonic particles, the use of high-performance metallic nanoheaters may lead to: (i) the reduction of nanostructure concentration use; and (ii) the reduction of light fluence.

On selecting metallic nanoparticles for thermal applications, structures with a high absorption cross-section ($\sigma_{abs}$) are desired. High $\sigma_{abs}$ values may lead to significant light energy absorption, and therefore increasing the NP temperature. In general, by increasing the size of plasmonic particles, not only the absorption cross-section is enhanced, but the scattering cross-section ($\sigma_{sca}$) value also rises. In that case, light energy is strongly scatter to the NP surround medium. Therefore, the Photothermal Conversion Efficiency (η) is known as an important figure of merit (FoM) on the evaluation of optical heating performance of nanparticles [24,25]. The Photothermal Conversion Efficiency is defined as the ratio of absorption cross section to extinction cross section ($\sigma_{ext}$), i.e.,:(2)$$\eta =\frac{\sigma_{abs}}{\sigma_{ext}}=\frac{\sigma_{abs}}{\sigma_{sca}+\sigma_{abs}}\;.$$

The η value (Equation (Equation 2)) quantifies the portion of the incident EM field absorbed by the nanostructure. Therefore, usually, small plasmonic nanoparticles present efficient coupling of light by absorbing the incident light with barely any scattering. As the particle grows bigger, both the absorption and scattering increase. However, for larger particles, the scattering process starts to be determinant on the light–NP interaction. The ability of an NP to lose heat to its surroundings is related to its surface area and volume. To grant efficient heat loss, the volume must be minimized, while the surface area must be maximized. Thus, NP morphology becomes relevant to thermoplamonics applications. Thus, Lalisse et al. described an effective way to evaluate the ability of a NP to generate heat, the Joule number (Jo), which is given by [26]:(3)$$Jo=\frac{\lambda_{ref}}{2\pi }\left(\frac{\sigma_{abs}}{V_{np}}\right)\;,$$
where $\lambda_{ref}\approx 1240$ nm is the reference wavelength of a photon with energy of 1 eV. Alternatively, Yakunin et al. explored the Arrhenius damage function along with gold NP size to assess the efficacy of local hyperthermia, showing that the NP absorption efficiency may not be enough to determine photothermal damage of biological tissues [27]. Moreover, Morales-Dalmau et al. show that cellular uptake of AuNRs and temperature rise in photothermal heating are highly dependent on NP shape and size [28].

The plasmonic heating induced by pulsed lasers and CW sources are governed by distinct dynamics, and therefore, the features of the excitation source should also be considered to identify efficient nanoheaters. Under fs pulse illumination, the pulse duration is roughly 10 times shorter than the electron–phonon relaxation time. Therefore, if the pulse repetition period is longer than $\tau_{d}$, the power delivered by the pulse is absorbed before the energy starts to be transferred from the hot-electron cloud to the NP lattice [29]. Considering a uniform temperature distribution inside the NP, the change of temperature over time ($\Delta T_{np}^{fs}$) is driven by fs laser absorption and heat conduction to the surrounding medium over a timescale $\tau_{d}$. Therefore, the temperature behavior of the nanoparticle may be expressed by Equation (S4). Considering that no phase transitions are involved in the heating process, Equation (S4) yields a simple analytical solution, which can be written as: (4)$$\Delta T_{np}^{fs}\left(t\right)=\frac{\langle I\rangle }{\rho_{Au}c_{pAu}f\left(1-\tau_{ep}/\tau_{d}\right)}\;\frac{\sigma_{abs}}{V_{np}}\;\left[e^{-t/\tau_{d}}-e^{-t/\tau_{ep}}\right].$$

Here, 〈I〉 denotes the average pulse intensity, *f* is the laser repetition rate and $V_{np}$ is the NP volume ($V_{np}=\frac{4}{3}\pi R_{eq}^{3}$). Under pulsed illumination, the maximum temperature change is not only localized in space, but also in time, as the heat front diffuses away from the NP surface. The temperature variation is directly proportional to the ratio of its absorption cross section to volume ($\Delta T_{np}^{fs}\propto \sigma_{abs}/V_{np}$). By comparing Equations (Equation 3) and (Equation 4), one can observe that $\Delta T_{np}^{fs}\propto Jo$. Hence, Jo presents itself as a good FoM to assess temperature variation in single NPs induced by fs pulses. Moreover, depending on NP size, this can be extended to pulses up to tens of ns, regarding that most of the pulse energy is absorbed before thermal diffusion becomes effective.

If a train of fs pulses is used to induce photothermal conversion in a NP, the temporal profile described by Equation (Equation 4) is replicated after each pulse. If the laser repetition period (1/f) is longer than $\tau_{d}$, there is no cumulative heating in the NP initially. However, since the thermal diffusivity of water is much bigger than gold, the heat diffuses slower in water and temperature “accumulates” in the surrounding medium. Although such behavior is insufficient to support cumulative heating from a single nanostructure perspective, the superposition of heat generated by many nanoheaters favors large temperature changes in macroscopic media [30].

For continuous excitation, energy absorption and conductive transport of heat to the surrounding medium happens concurrently, leading to lower rises of the NP temperature. The morphology of a nanostructure plays an important role in they way in which heat is exchanged with its surroundings. The ability to lose heat is related to the surface area of the NP, and thus, a larger surface area allows for more efficient conduction of heat outwards. Considering this, Baffou et al. generalized the expression for the steady-state maximum temperature change that an NP of arbitrary shape can reach [31]:(5)$$\Delta T_{np}^{CW}=\frac{I}{4\pi \kappa_{m}}\frac{\sigma_{abs}}{R_{eq}\beta }\;,$$
in which β is the shape-correction factor. For gold nanorods (AuNR), β is a function of the NR aspect ratio (*AR = L/D*) and is given by $1+0.096587\;ln^{2}\left(AR\right)$ [31]. The shape factor β, the absorption cross section and the equivalent radius are all a function of NP shape. Therefore, changes of the NP dimensions may increase or reduce the steady-state temperature that the nanostrucuture can reach.

In this work, we introduced a methodology based on the analysis of important thermo-optical figures of merit to select high performance nanoheaters. We considered the ratio $\sigma_{abs}/R_{eq}\beta$, named Steady-State Factor ($S^{2}F$), as a new figure of merit to evaluate the steady-state heating of plasmonic particles under continuous illumination. A rational methodology, based on the analysis of relevant FoM, on identifying high-performance plasmonic nanoheaters is explored. As a case study, the optimization of gold nanorods (AuNRs) in water for heat generation was performed appraising η, Jo and $S^{2}F$ under an NP size-dependence approach. Moreover, collective heating of colloidal NR was also evaluated considering a theoretical and experimental approach.

## 2. Method and Materials

### 2.1. Nanorod Optimization

Optical heating of metallic nanostructures is associated to Joule losses, which can be understood in a semiclassical approach as the result of the collisions between conduction electrons and ions of the gold crystal lattice, represented by the damping factor on the Drude model. As the size of the metallic structure decreases, the collisions between conduction electrons and NP surface increase, which becomes more relevant for nanostructures with dimensions comparable to the mean free path of the conduction electrons. This aspect leads to an increase of the damping factor, causing the permittivity of metallic NPs to become size dependent [32]. For NPs smaller than 5 nm, however, quantum effects related to a reduced number of atoms in the crystal start to appear [33], and the size-dependent model is no longer valid. The size-dependent permittivity of metallic NPs is given by [34]:(6)$$\epsilon \left(\omega ,L_{eff}\right)=\epsilon_{bulk}\left(\omega \right)+\frac{\omega_{p}^{2}}{\omega^{2}+j\omega \gamma_{0}}-\frac{\omega_{p}^{2}}{\omega^{2}+j\omega \left(\gamma_{0}+A\frac{v_{F}}{L_{eff}}\right)}\;.$$
where ω is the angular frequency of incident light. The frequency-dependent permittivity of bulk metals ($\epsilon_{bulk}$) accounts for intraband and interband contributions in the spectral region of interest. In this work, the bulk gold permittivity provided by Johnson and Christy was used [35]. $L_{eff}$ is the effective mean free path of conduction electrons. For convex shapes, $L_{eff}=4V_{np}/S$ [34] and *S* is the surface area of the NP. The parameter *A* describes the scattering interaction at the surface of the nanostructure. The surface scattering parameter for nanorods (NRs) lies between 0.25 [36] and 0.5 [37]. Table 1 presents the values used in this work for each gold parameter present in Equation (Equation 6).

Plasmonic properties were appraised by means of Finite Element Method (FEM) electromagnetic simulations in COMSOL Multiphysics, in which a single AuNR of length *L* and diameter *D* (Figure 2a) was placed in a dielectric medium (water). The NR longitudinal axis was aligned with the polarization of incident light. The surrounding medium refractive index was assumed to be wavelength independent and bounded by a perfectly matched layer (PML) with spherical symmetry, mimicking an open boundary and avoiding the reflection of scattered light. NR length *L* was swept from 15 nm up to 200 nm for a set AR. The process was extended for various AR values. The results obtained from FEM simulations were validated by comparison with Mie–Gans theory, using few NR structures with different AR [40] (see Appendix A).

The proposed methodology is limited to the analysis of NPs that does not present radiative processes. The FoMs may be extended for non-metallic nanoheaters, such as semiconductor nanoparticles and other materials that present radiative damping by taking into account its quantum efficiency. This is beyond the scope of this work and will not be discussed.

### 2.2. Experimental Temperature Evaluation

Gold nanorod samples were chosen to evaluate the collective heating of a colloid under 800 nm laser irradiation. CTAB-stabilized colloidal samples of AuNRs, with sizes of 41 × 10 nm, 90 × 25 nm and 134 × 40 nm, in deionized water were acquired from Nanopartz (Loveland, CO, USA). Since NP optimization was performed appraising thermo-optical parameters of single AuNRs, all samples were diluted in distilled water, leading to samples with the same NP volumetric density (10^15^ m^−3^). UV-Vis absorbance spectroscopy (Ocean Optics USB2000, Orlando, FL, USA) was performed before and after irradiation of each sample to ensure that the temperature assessment did not degraded the colloid.

Sample irradiation was performed in a 2 mm cuvette with a 10 mm optical path by a tunable fs laser (Coherent Chameleon Vision II, 140 fs/80 MHz). A cylindrical lens was used to produce a light sheet profile capable of illuminating a large area of the specimen placed in the cuvette (400 μL). A thermal imaging camera (Flir E4, Wilsonville, Oregon, USA) was then used to acquire the temperature evolution of the specimen in real time. The experiment comprised 20 min of laser heating (1.5 W@800 nm), followed by 20 min of cooling (no light on the sample). During the process, images of the temperature distribution in the cuvette were taken every 5 min. For each sample, the measurements were performed in triplicate.

## 3. Results and Discussion

Following the proposed optimization procedure, the thermo-optical properties of the AuNR with various lengths and diameters were depicted, as shown in Figure 2. The colormap in Figure 2b delineates LSPR peak position ($\lambda_{p}$) as a function of NR length and diameter. The dashed and dash-dotted lines in the colormap represent the AuNR sizes in which $\lambda_{p}$ occurs for ∼800 nm and ∼1064 nm, respectively. The wavelengths 800 nm and 1064 nm correspond to important laser lines within the first and the second biological NIR transparency window. By changing the particle size, keeping AR constant, the LSPR peak barely moves for NPs with D smaller than 25 nm. However, as the volume of the NR increases, dephasing effects on the conduction electrons start to become relevant and the LSPR peak is red-shifted [41]. Therefore, for bigger NRs, the dashed and dash-dotted lines start to bend.

Figure 2c shows the colormap for the Photothermal Conversion Efficiency of AuNR. Large values of η are obtained for NR with D smaller than 50 nm. Nevertheless, η values becomes smaller by increasing NR volumes, the overall absorption and scattering cross-sections rise, and $\sigma_{sca}$ overcomes $\sigma_{abs}$. The dashed and dash-dotted curves in the colormap indicate AuNR dimensions in which the plasmon peak occurs at 800 nm and 1064 nm, respectively. Figure 2d depicts the decreasing behavior of η as the size of the NP, with resonance at 800 nm and 1064 nm, is increased. Therefore, it is compelling to think that smaller-size particles are preferable for heat generation.

In order to obtain the AuNR diameter, we must explore the colormaps in conjunction with the FoM plots as a function of NR length. Figure 2d,f,g allows the identification of the AuNR length associated with the highest FoM vales. In order to obtain the AuNR diameter, the colormaps in Figure 2c,e,g must be explored. Notice that the dashed-dotted and the dashed lines in the colormaps corresponds to the ordered pair (diameter, length) in which all the AuNRs have LSPR fixed in the same wavelength (800 nm for the dashed line and 1064 nm for the dashed-dotted line). Hence, for a fixed wavelength, one NR length is related to only one NR diameter.

Figure 2e is the Joule number colormap for AuNRs in water. One can observed that particles with a diameter smaller than 30 nm show high Jo values. As stated previously, the capacity of an NP to convert light into heat is proportional to $\sigma_{abs}$ and inversely related to its volume. As the particle grows bigger, so do both contributions. However, $\sigma_{abs}$ and volume grow at different rates, which leads to a reduction of Jo values as the NR increases. Figure 2f outlines Jo values as function of AuNR length in water for the traces at which $\lambda_{p}$ are 800 nm and 1064 nm. The Jo values follow a trend similar to η, where smaller NP sizes are more efficient for photothermal conversion.

The indication of maximum heat generation for long periods of illumination is exerted by the Steady-State Factor ($S^{2}F$). Figure 2g presents the $S^{2}F$ colormap for AuNRs in water, while Figure 2h outlines $S^{2}F$ values as function of AuNR length for the inset traces. The red-color area in Figure 2g indicates that high $S^{2}F$ values are obtained for particles of long length (bigger than 60 nm) and short diameter (smaller than 40 nm). On defining the $S^{2}F$ factor, the NP size dependency is represented by the equivalent radius, while shape dependency is introduced by the shape factor β. The dashed and dashed-dotted lines in Figure 2h show that the best AuNR for single-particle CW photothermal conversion at 800 nm and 1064 nm in water are, respectively, ∼90 × 25 nm and ∼150 × 30 nm.

Our results show that $S^{2}F$ values tend to peak. Differently to the Jo, where smaller NRs are better, $S^{2}F$ maximum occurs at longer AuNR lengths (and diameters), indicating that on engineering plasmonic nanoheaters, it is important to consider the illumination regime (pulsed or CW). Furthermore, optimal AuNR identification unveiled FoM variations up to 700% when considering structures with the same peak wavelength.

Each NP shape will provide a different behavior (values) to Jo and $S^{2}F$. However, several applications that rely on photothermal conversion demand a temperature increase at the macroscopic scale. It is, therefore, relevant to investigate the effect of the NP optimization under collective heating. Considering a medium containing multiple plasmonic NPs exited by a laser (pulsed or CW), the total global temperature change is given by the superposition of each individual NP in the steady state, i.e., [30]:(7)$$\Delta T_{global}=\sum_{n=1}^{{\#}_{np}}\Delta T_{np}^{\left(n\right)},$$
in which ${\#}_{np}$ is the total number of NPs and *n* is the *n*th NP of the sample. By supposing all NPs’ heat, the thermal contribution is the same for each NP and the summation can be replaced by a volume integral over the illuminated region ($V^{\prime }$) with constant volumetric density of NPs ($C_{np}$), as (Appendix A
):(8)$$\Delta T_{global}^{CW}=C_{np}\Delta T_{np}^{CW}R_{eq}\left(\int_{V}\;\frac{dV^{\prime }}{r^{\prime }}\right)$$
and
(9)$$\Delta T_{global}^{fs}=C_{np}\langle \Delta T_{np}^{fs}\rangle V_{np}\left(\frac{3}{4\pi }\int_{V}\frac{dV^{\prime }}{r^{\prime {\;}^{3}}}\right)\;,$$
where $\langle \Delta T_{np}^{fs}\rangle$ is the average temperature of the NP between two consecutive non-cumulative fs laser pulses ($\langle \Delta T_{np}^{fs}\rangle =f\int_{t}^{t+1/f}\Delta T_{np}^{fs}\left(t^{\prime }\right)dt^{\prime }$).

Qualitatively, the global temperature change is proportional to $C_{np}\Delta T_{np}$, which, in turn, is proportional to $C_{np}\sigma_{abs}$ in both cases. For thick or very concentrated samples (large $C_{np}$), the analysis of all previously discussed FoMs become irrelevant, since most of the energy delivered stays in the sample. In such case, as long as the experimental conditions remain the same, all samples must experience the same global temperature change.

Figure 3a depicts the experimental setup employed to perform temperature rise measurements. The laser sheet allowed the sample temperature to change uniformly. Global temperature evolution was assessed taking the average temperature change throughout a rectangular region in the sample. The uniform temperature increase of the specimens can be verified on the thermographic stills of Figure 3b. In it, the specimen location is easily detectable due to the quasi-uniform temperature distribution inside the region of interest (rectangle area).

The time evolution of global temperature was acquired for each sample. The results show that the 134 × 40 nm sample enabled higher temperature change among the appraised samples. The measured global temperature change was 8.64 ± 0.12 °C, 12.36 ± 0.14 °C and 14.60 ± 0.24 °C for AuNRs of 41 × 10 nm, 90 × 25 nm and 134 × 40 nm, respectively. When the same amount of gold is the same for different specimens of distinct AuNR sizes, the final global temperature weighed by the total Au mass becomes proportional to the Joule number ($\Delta T_{global}/M_{tot}\propto Jo$). The experimentally appraised global temperatures were contrasted with the total mass of gold present in each specimen, and the result is displayed in Figure 4a, where temperature change per gold mass tends to follow the Joule number. Similarly, by dividing the total temperature rise by the total number of NPs in each sample, the temperature rise tends to follow the gold nanorod absorption cross-section ($\Delta T_{global}/NP\;\propto \;\sigma_{abs}$), as shown in Figure 4b. Therefore, the choice for the best gold nanorod size for temperature rise under collective heating depends on the circumstances delineated by the application. If the same mass of material is considered, the FoM of choice is Jo. However, if the application concerns the same number of NPs, the NP absorption cross-section must be maximized, while collective heating mediated by plasmonic NPs is maximized by maximizing the NP absorption cross-section.

## 4. Conclusions

We established a computational framework for the optimization of metallic NPs for heat generation. As a case study, the optimal dimensions of AuNRs were obtained for both non-cumulative fs pulses and CW excitation. Figures of merit were identified and described for the optimization of heat generation: (i) the Joule number (Jo) for non-cumulative short pulses and (ii) the Steady-State Factor ($S^{2}F$) for continuous irradiation, adequate for most metallic NP shapes. In particular, Jo and η values show a similar trend, indicating that smaller NP sizes are more efficient for photothermal conversion under short pulse excitation. The evaluated figures of merit present substantial potentialities and may be applied to optimize nanoheaters other than AuNRs regarding applications suitable for a single or few NPs, such as photoporation and ultrafast thermal switching. The effect of such optimizations in collective heating was appraised and its application in macroscopic heating was discussed. The experimental results show that if the same total mass of gold is used, the collective heating follows the trend of the Joule number. Even though the demonstration of the described methodology is restricted to AuNRs in this work, the proposed analysis is adequate for different nanoparticle shapes and material compositions.

## Figures and Tables

**Figure 1 nanomaterials-12-04188-f001:**
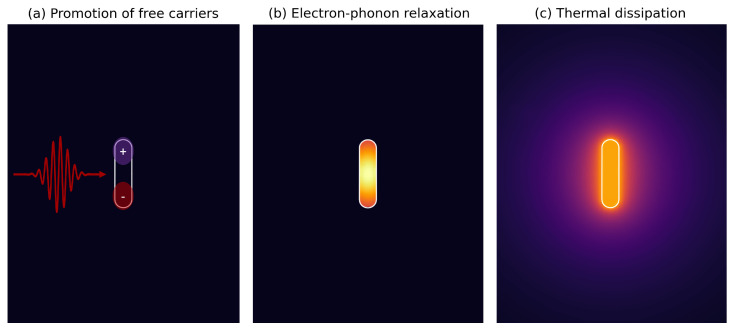
Illustration of photothermal phenomena dynamics of metallic NPs, from (**a**) promotion of free carriers (fs) and (**b**) electron−phonon relaxation (ps), to (**c**) heat exchange with the surrounding medium (ps to ns).

**Figure 2 nanomaterials-12-04188-f002:**
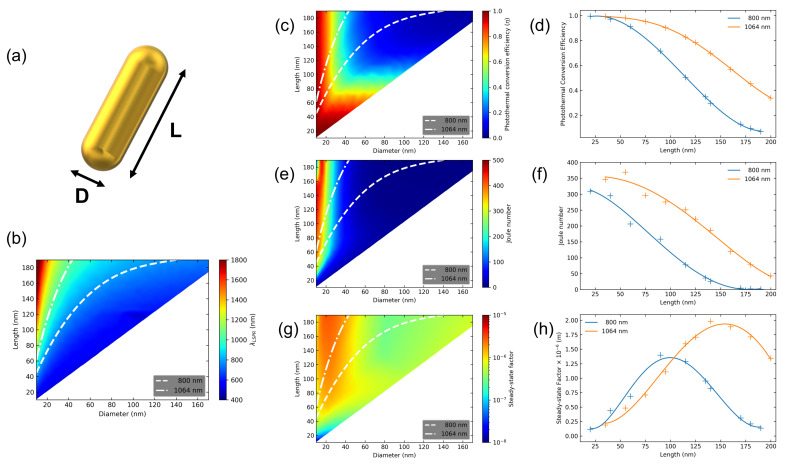
(**a**) Scheme of an AuNR and its dimensions. (**b**) LSPR colormap as a function of AuNR length and diameter. (**c**,**e**,**g**) Colormaps for η, Jo and $S^{2}F$, respectively, for various AuNR dimensions. The outlined dashed and dash-dotted curves in all colormaps depict AuNR dimensions in which the plasmon peak occurs for 800 nm and 1064 nm, respectively. (**d**,**f**,**h**) Evolution of η, Jo and $S^{2}F$ as a function of AuNR size. The blue curves represents AuNR sizes with LSPR in 800 nm, while the orange curves portrays AuNR sizes with LSPR in 1064 nm.

**Figure 3 nanomaterials-12-04188-f003:**
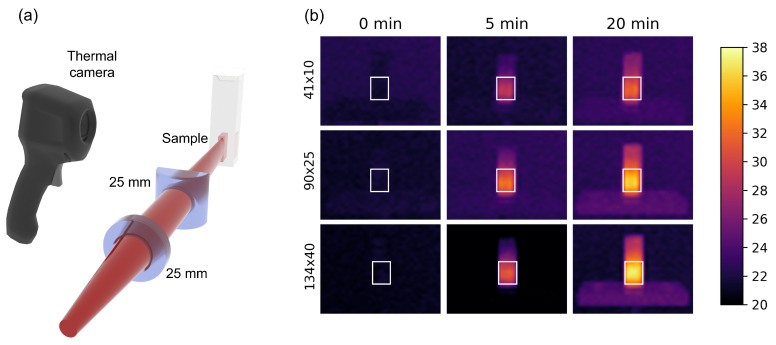
(**a**) Experimental details picturing the light sheet optics and the positioning for temperature acquisition. (**b**) Thermographic stills of the specimens during laser irradiation for temperature distributions in the AuNR samples and surroundings after 0, 5 and 20 min. The rectangles delineate the specimen location in the cuvette. The colorbar (virtual) restricts the range of temperature change in all samples.

**Figure 4 nanomaterials-12-04188-f004:**
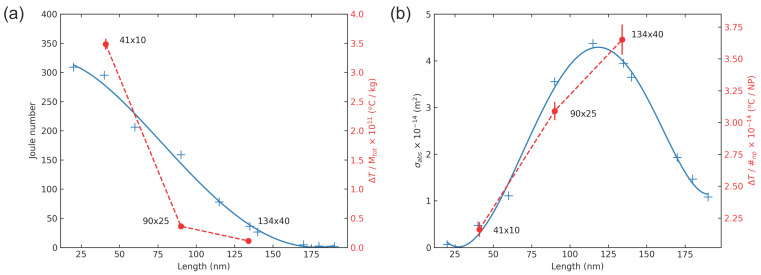
(**a**) Measured global temperature change divided by the total mass of gold of each sample (red dashed line). For the same amount of gold, the total temperature rise follows the Joule number (blue line). (**b**) Measured global temperature change divided by the total number of NPs in each sample (red dashed line). For the same number of NPs, the total temperature rise follows the absorption cross-section (blue line).

**Table 1 nanomaterials-12-04188-t001:** Bulk parameters for AuNRs.

Parameter	Value	Description	Reference
$\omega_{p}$	$1.369\times 10^{16}$ rad/s	Plasmon frequency	[38]
A	0.25	Surface scattering constant	[36]
$v_{F}$	$1.4\times 10^{6}$ m/s	Fermi velocity	[39]
$\gamma_{0}$	$1.07\times 10^{14}$ s${}^{-1}$	Bulk damping factor	[38]

## Data Availability

The data presented in this study are available on request from the corresponding author.

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
