# Peer review of "Selecting High-Performance Gold Nanorods for Photothermal Conversion"

_nanomaterials, 2022, doi:10.3390/nano12234188_

Round 1

Reviewer 1 Report

This manuscript reports the study of methods for selecting gold nanorods based on their potential efficacy in plasmonic photothermal therapy applications. The authors outline the potentially useful application of two different yet related figures-of-merit for assessing gold nanorod performance and provide a combination of theoretical analysis and computation and empirical results indicating the effectiveness of their approach. The manuscript is generally succinctly and clearly written and supported by an appropriate set of figures summarising the approach. There are some points that the authors could address and these are detailed below.

1.       Page 2 line 37 Change “. . . reelected to the surrounding medium . . .” to “. . . released to the surrounding medium . . .”

2.       Page 5 lines 164-167 “NR length . . . different AR [40].” In Figure 2 I can see the results for your sweep of lengths from 15-200 nm with variation of the aspect ratio (AR) but you mention validation by comparison to Mie-Gans theory at discrete different aspect ratios and I cannot see where this data comparison is made. Please clarify this and provide details on these data.

3.       Page 5 line 171 “. . . DIH . . .” This is a non-standard way of abbreviating deionized water and it would be better to write it out in full instead.

4.       Figure 2 legend. Whilst the impact of changing aspect ratio is fairly clear from Figures 2b, c, e and g, it is not clear to me how you have settled on particular diameters of the nanorods for Figures 2d, f, h or what those diameters are. Clarify this in the manuscript.

5.       Page 7 lines 220-221 “. . . Figure 2(h) shows . . . ~150x30nm.” Related to comment 4 above, how did you arrive at 25 nm and 30 nm diameters exactly? This is not clear from Figure 2h. Clarify this in the manuscript.

6.       Figure 4 and Figures S3-5. Whilst the three measurements from one sample given in Figures S1-3 are useful, I would like to see some clearer evidence of experimental repeatability for these data including in Figure 4, e.g. standard errors of measurements, error bars, evidence of replication across multiple samples and associated assessment of variability.

Reviewer 2 Report

The main goal of this paper is slightly unclear for me. Next, many ammount of the mathematical equations not increase the clarifity of the manuscript. These type elements should be presented in the supplementary material. Paragraph 4 include generally a constatation of some experimental obervation. So, due to issues mentioned above, the manuscript should be reorganised.

Additional comment:
Page 7, line 213
What mathematical apparatus was used in the optimization process? What goal function? What baseline data? The entire optimization process requires a precise explanation in line with the definition of the term "optimization".

Reviewer 3 Report

Dear author,

After the review process, I have several comments: if it is an article, it should contain numerical data in the abstract; in figure 1 legend, for example, should be included how was realized; the limitation of the study should be mentioned and future valorization; the paper is more proper as a communication rather than an article.

Best regards!

Round 2

Reviewer 1 Report

The authors have made appropriate changes to the manuscript in response to my earlier review comments.